# Dynamic control of DNA condensation

Siddharth Agarwal [1,2], Dino Osmanovic[1], Mahdi Dizani [1], Melissa A. Klocke[1,3] & Elisa Franco [1,2] ✉

Artificial biomolecular condensates are emerging as a versatile approach to organize molecular targets and reactions without the need for lipid membranes. Here we ask whether the temporal response of artificial condensates can be controlled via designed chemical reactions. We address this general question by considering a model problem in which a phase separating component participates in reactions that dynamically activate or deactivate its ability to self-attract. Through a theoretical model we illustrate the transient and equilibrium effects of reactions, linking condensate response and reaction parameters. We experimentally realize our model problem using star-shaped DNA motifs known as nanostars to generate condensates, and we take advantage of strand invasion and displacement reactions to kinetically control the capacity of nanostars to interact. We demonstrate reversible dissolution and growth of DNA condensates in the presence of specific DNA inputs, and we characterize the role of toehold domains, nanostar size, and nanostar valency. Our results will support the development of artificial biomolecular condensates that can adapt to environmental changes with prescribed temporal dynamics.

The discovery of cellular organelles without a membrane is shifting the paradigms of our understanding of life and its origins[1]. These organelles form as molecules spontaneously condense into phase-separated compartments. A growing number of studies aims at elucidating the design principles underlying condensation[2] with the goal of engineering biomolecules that can isolate components and reactions by design, with major implications in biology, biomaterials science, and medicine.

A central question in this context is how biological organisms manage to control the appearance and disappearance of condensate in non-equilibrium conditions. While it is well-known that temperature and ion levels control phase transitions, cells must use other mechanisms to grow and dissolve organelles over time. Stoichiometry, for example, influences the emergence of multi-component biological condensates[3,4], suggesting an important role for chemical reactions that control the abundance of species participating in the condensate. These chemical reactions may directly regulate the production or degradation of participating species or supply chemical agents that can disrupt the multivalent bonds necessary for separation to occur, leading to reversible growth and dissolution of condensates. While it is

ideal to test these hypotheses by selectively engineering natural condensates[5], simplified minimal systems can provide useful biophysical insights while circumventing the native cellular complexity[3,4,6–8]. These systems should offer the possibility to finely control both specific and non-specific interactions among the phase-separating components and the kinetics of chemical reactions that regulate their properties.

DNA nanotechnology is an ideal platform to explore questions pertaining to how chemical reactions can regulate condensate formation. This field has demonstrated that DNA and RNA molecules can be designed to implement chemical reaction networks, self-assembling structures, and amorphous condensates[9,10]. In all these systems, the interactions among nucleic acid domains are prescribed by Watson–Crick–Franklin base-pairing rules, and sequences are designed using computer algorithms[11,12]. DNA structures and complex reactions can include hundreds to thousands of single strands whose programmed domain-level interactions yield the overall desired operation[9]. In addition to designed thermodynamic interactions, DNA nanotechnology also offers methods to control kinetic responses with precision, by taking advantage of strand displacement and branch

[1]Mechanical and Aerospace Engineering, University of California at Los Angeles, Los Angeles, CA 90095, USA. [2]Bioengineering, University of California at Los Angeles, Los Angeles, CA 90095, USA. [3]Deceased: Melissa A. Klocke. ✉e-mail: efranco@seas.ucla.edu

migration reactions[13]. This coherent design and implementation platform enabled the demonstration of systems in which kinetic reactions and self-assembling structures coexist and exchange information for sustained periods of time[14–18]. These approaches can be readily extended to couple DNA-based reactions with DNA-based condensates.

Here, through theory and experiments we describe a system of condensate droplets whose formation is temporally, reversibly controlled by chemical reactions that activate and deactivate the subunits participating in the condensate. First, we develop a mean field theory of condensates in the presence of reactants that affect their ability to phase separate, and show how the impact of said chemicals leads to phase diagrams for the system which have similar features to those introduced by changes in temperature. We then apply this theoretical approach to an experimental system in which DNA condensates arise from the sequence-specific interactions of star-shaped DNA motifs known as nanostars[10,19,20]. Using principles from DNA nanotechnology, we demonstrate that droplets can be reversibly grown and dissolved through sequence-specific activation and deactivation of the nanostar bonds that reduce the nanostar valency. We measure the temporal evolution of droplets size and number under different designs of DNA motifs and reactions, elucidating the dependence of out of equilibrium condensate behavior on various factors under our direction, such as the concentrations of the components involved, their binding affinity, the valency of the nanostars and the size of the nanostars. In particular, we highlight that monomer valency reduction is an efficient mechanism to control condensate formation, when compared to simply regulating the total concentration of subunits. Our results provide a versatile, minimal DNA-based toolkit for designing and controlling dynamically responsive macroscopic condensates, which may be used to explore fundamental questions in biomolecular phase separation science. This toolset will also facilitate the synthesis of biomolecular materials in which the formation of condensates can be temporally and autonomously controlled via a variety of nucleic acid-compatible chemical reactions[21].

## Results

### A theoretical model suggests means to control condensation through biochemical reactions

To formalize the notion of chemical control of phase separation, we consider an ideal, simplified case in which a single biomolecular species $n$ undergoes phase separation. This process is driven primarily by the species concentration and by the temperature of the system, as illustrated by well-understood phase diagrams[22]. However, it is less clear how temporal changes of the concentration of $n$ introduced by chemical reactions that deactivate or activate $n$ will affect the formation and the kinetics of phase separated condensates, despite recent theoretical work in this area[23,24].

To investigate this problem, we built a mean field model of molecular condensation from the Cahn-Hilliard equation for phase separation in 3D, with the addition of mass action chemical kinetics that introduce temporal changes in the amount of monomers of phase separating species[25]. We introduce elementary chemical reactions that convert the monomers between their active form $n$, which phase separates, and their inactive form $n_0$, which cannot: an inhibitor molecule $I$ mediates the deactivation of monomers, while an activator mediates their reactivation by removing inhibitor, a process that generates waste $W$:

$$n + I \underset{k_b}{\overset{k_f}{\rightleftharpoons}} n_0 \tag{1}$$

$$n_o + A \underset{h_b}{\overset{h_f}{\rightleftharpoons}} n + W \tag{2}$$

The full derivation of the mean field equations, along with a description of their computational integration, is provided in the Supplementary Note 1 in the Supplementary Information file. Using these model equations and linear stability analysis, it is possible to derive phase diagrams that correspond to controlling phase separation using chemical reactions. The phase diagram informs us about how the addition of inhibitors modifies the ability of the system to phase separate (Fig. 1 A and Supplementary Fig. 17), as the on ($k_f$) and off ($k_b$) parameters of the inhibition reaction are changed. It can be seen that if these parameters are comparable, the effect of adding inhibitors is similar to raising temperature in a classical phase separating system[23]. However, in contrast to raising temperature, the effect of changing inhibitor concentration has targeted effects only on a specific species. In addition, these diagrams allow us to distinguish between two modes of droplet dissolution. When $k_f = k_b$ in Fig. 1A, by supplying inhibitors we can move the system across the phase boundary, from the region where phase separation occurs (orange) to the region where it does not (gray), causing condensate dissolution. This dissolution would proceed *thermodynamically*, as condensates would not be stable with respect to a dispersed phase. Adding an amount of inhibitor that does not bring the system outside of the orange region might lead to an initial *kinetic* decrease in condensate size, but at longer time-scales condensates would regrow via coarsening. This dichotomy between kinetic and thermodynamic dissolution implies that, depending on the phase boundary, one may not need a 1:1 ratio between phase separating monomer and inhibitor to observe complete condensate dissolution.

Further clarity about these processes can be gained by performing computational simulations exploring the system's kinetic response to the introduction of inhibitor and activator. Figure 1B illustrates the behavior of a system in which droplets are allowed to phase separate first, with later sequential addition of inhibitor and activator at different ratios (0.25×, 0.5×, and 1× the level of monomer). These computationally generated images qualitatively confirm that addition of inhibitor causes droplet dissolution, and addition of activator promotes their regrowth. From these images, we gathered quantitative insight on the evolution of droplet size over time (normalized relative to the forward binding rate and the monomer concentration). At low concentrations of inhibitor (0.25×), its effect saturates over time, indicating that there is not enough inhibitor to move the system into a regime where the droplets are not thermodynamically stable, so they may eventually regrow. Using a different parameter set and performing longer simulations, in Supplementary Fig. 18 we exemplify a case in which regrowth does occur. (Note that jumps in the average trajectories are due to changes in the discrete number of droplets being simulated.)

Consistent with expectation, the average droplet size decreases faster when the inhibitor to monomer ratio is larger (Fig. 1C). This process is however affected by the inhibitor diffusion rate $D$, which manifests as a change in the temporal scaling law when $D$ is between 1 and 10 (Fig. 1D). This indicates that inhibition may be surface-driven (slower diffusion), or volume driven (faster diffusion). Visually, surface-driven dissolution would result in droplets shrinking while retaining their shape, in contrast with blurred droplets occurring under volume-driven dissolution (Supplementary Fig. 19). Overall, increasing either the inhibitor concentration or its diffusion coefficient promotes faster droplet dissolution, though apparently with different scalings.

The addition of activator (in stoichiometric amount to the inhibitor) to a mixture of $n$ and $n_O$ promotes droplet regrowth. Figure 1E shows that the speed of regrowth depends on both the initial droplet size and on the activator to monomer ratio. If we focus on the average droplet volume, we note a rapid increase at 0.25× and 0.5× ratios; in contrast, a slow increase is noticeable at a 1× ratio. This non-monotonic behavior emerges due to the presence of three processes driving growth: monomer activation, droplet nucleation, and droplet

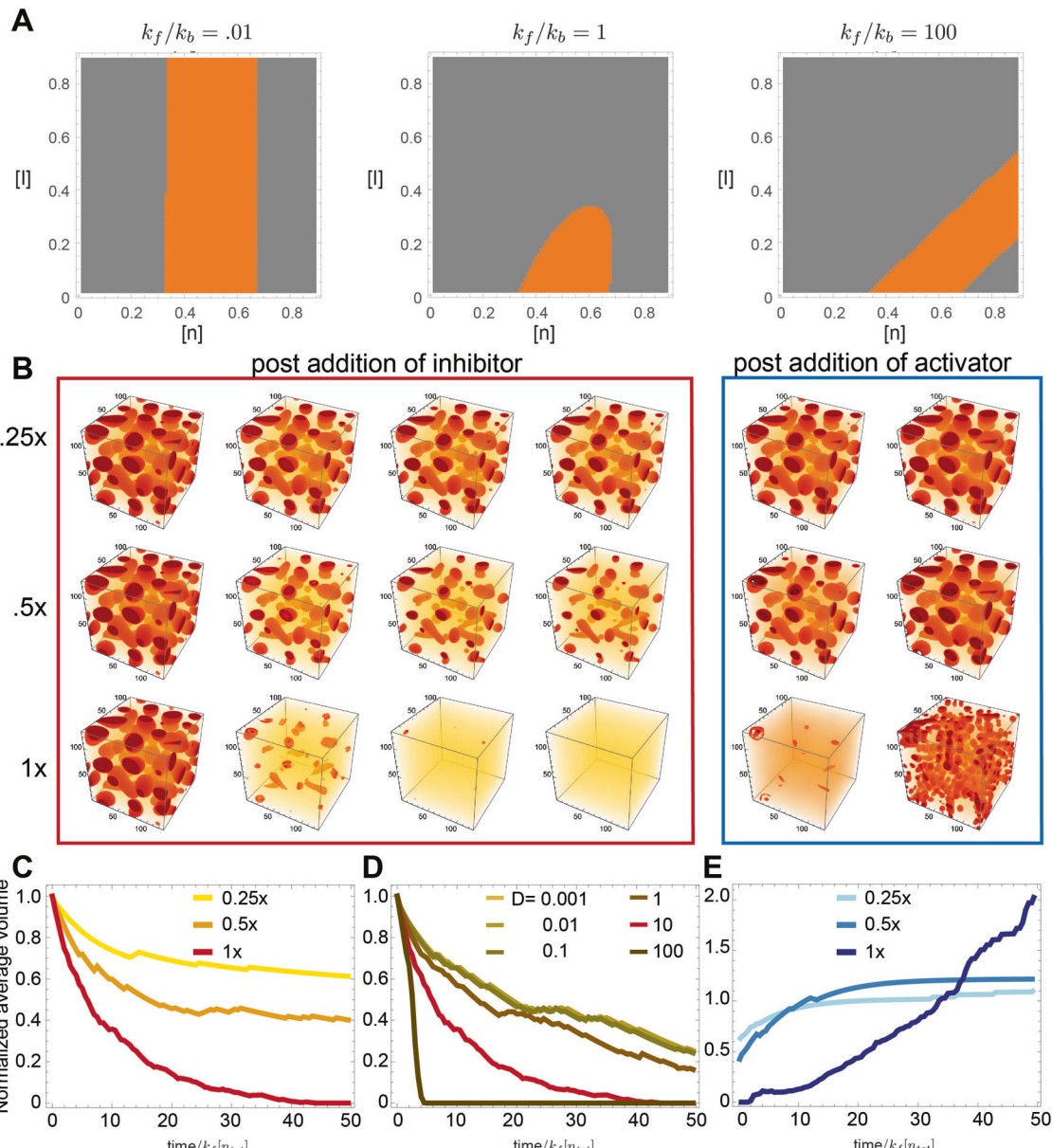

**Fig. 1 | A computational model illustrates the effects of inhibition and activation chemical reactions on phase separated droplets. A** The effect of adding an inhibition reaction on the phase diagram depends on the ratio of the forward and backward rate constants; areas in orange correspond to two coexisting phases. When the rate constants are comparable, increasing the inhibitor concentration has an effect akin to increasing temperature. **B** Computationally generated images of droplets. We combined the Cahn–Hilliard equation for phase separation with chemical reactions: an inhibitor deactivates the phase separating material, causing droplets to shrink over time; droplets regrow after addition of activator. **C** Average droplet area over time, at different inhibitor concentration. **D** Average droplet area over time as an inhibitor species with different diffusion constants is supplied to the system. We identify a slow regime (surface driven) in which as the surface decreases in time, the time taken for the droplet to dissolve will increase, leading to a slowing down of dissolution in time. The opposite is the case for inhibitors diffusing fast. **E** Average droplet area after addition of different amounts of activator. Regrowth speed does not depend on the diffusion constant of the activator (Supplementary Fig. 20). Simulation parameters are available in the Supplementary Note 1.

coarsening. When a large number of monomers are rapidly activated (Fig. 1B, post addition of 1× activator) nucleation of new, small droplets dominates over the coarsening of existing ones, resulting in slower increase of their mean size. As one would anticipate, the diffusion rate of the activator does not significantly influence droplet growth (Supplementary Fig. 20A). Finally, simulations in Supplementary Fig. 20B show that the half time of dissolution of individual droplets scales with a power law of the droplet radius (larger droplets should take longer to dissolve).

This simple theoretical model establishes a baseline expectation for the temporal behavior of condensates as monomers are activated and deactivated, and for the degree of controllability of the average condensate size. To test these expectations, we next develop an experimental platform to build dynamic biomolecular condensates using DNA nanotechnology.

## DNA condensate droplets can be controlled via strand invasion and displacement

We used synthetic DNA components to probe the effects of monomer activation and deactivation reactions on condensation. We adopted multiarm DNA motifs, or DNA nanostars, that interact via complementary palindromic 'sticky-end' (SE) domains present on the end of each arm[10,19,20]. We started our investigation with three-arm motifs (valency equal to three) characterized by Sato et al.[20], each arm having

a four-base sticky-end, shown in Fig. 2A. After assembly by thermal annealing the nanostars can be considered monomers that spontaneously yield DNA-rich droplets, which in our case are expected to remain liquid at room temperature (27 °C) while coarsening and coalescing into larger droplets[20]. Nanostars were annealed at 5 μM concentration unless otherwise noted. We modified one of the strands of the nanostars to include a fluorescent dye (Cy3), and we used microscopy images to gather statistics of the condensate droplet diameter as a function of time. Surface wetting and nucleation was avoided by using BSA coated slides (additional details about sample preparation, imaging, and image processing are in the Methods).

To control the growth and dissolution of droplets, we devised a system in which DNA species react with the nanostars, specifically to switch the state of their sticky ends between inactive and active (Fig. 2A, B). To enable the inactivation of nanostars, we designed strand invasion and displacement reactions controlling the availability of one of the sticky ends: we reasoned that a reduction of the valency from three to two would be sufficient to dissolve the DNA-dense droplets. To make it possible to deactivate one sticky end, we modified it to include a 'toehold' domain that is designed to have no secondary structure and to remain single-stranded. The toehold enables strand invasion by a DNA inhibitor molecule, henceforth dubbed 'invader', designed to be complementary to the toehold, the sticky-end sequence, and 6 nt of the nanostar arm. To allow for nanostar reactivation, the invader molecules are, in turn, designed to include their own toehold domain so that a complementary 'anti-invader' (AI) strand can displace the invader bound to a nanostar. This displacement reaction releases the sticky end of the nanostar, which recovers its original valency and its capacity to yield condensate droplets. Invader and anti-invader serve the purpose of activating and deactivating nanostar units via chemical reactions that are equivalent to those described in our theoretical model.

We first evaluated droplet dissolution upon the addition of invaders when the nanostars included toehold domains of different lengths. First, we verified that DNA nanostars modified to include a 7 nt toehold generate phase-separated droplets that grow over time due to coarsening and fusion events: Fig. 2C, left, shows a box plot of the droplet area, normalized with respect to the average area of one replicate measured at the beginning of the experiment (time 0 corresponds to 30 min after annealing, prior to adding invaders). The addition of a scramble invader sequence has no effect on the droplets (Supplementary Fig. 1). Invader was added to the condensate sample in a test tube, from which we extracted aliquots for imaging, as described in Methods. The addition of complementary 1× invaders to nanostars, including 7, 5, and 3 nt toehold domains, results in complete droplet dissolution within 5, 5, and 15 min, respectively (Fig. 2C, middle). This small change in dissolution speed is inconsistent with the fact that the velocity of invasion should scale exponentially with the toehold length[26]. A shift in the behavior of the droplets occurs when the toehold domain is eliminated (Fig. 2C, right): in this case, the droplet areas do not change significantly over our observation window, although there is a decrease in droplet number. We expect the invader to bind to unpaired sticky ends of the nanostars, most likely in the dispersed phase or on the surface of the droplets, overall slowing down droplet growth (the nanostar arm-invader interaction domain is the same as a nanostar–nanostar arm interaction). As it includes the palindromic sticky-end sequence, the invader competes for free sticky-ends and may be considered a "decoy" molecule. The competitive binding of invaders to nanostars may accelerate droplet dissolution in the 7, 5, and 3 nt toehold cases, reducing the kinetic difference one would expect.

Next, we tested the droplet behavior upon sequential addition of invader (7 nt toeholds) and anti-invader (6 nt toeholds) at an equimolar level relative to the nanostars. For example, fluorescence microscopy images of condensates before the invasion, after invasion, and after

anti-invasion are shown in Fig. 2D. While dissolution occurs in less than 2 min, droplet regrowth after the addition of anti-invader is a slow process driven by nucleation, coarsening, and coalescence of droplets. Unless otherwise noted, the invader and anti-invader strands used in the rest of the paper have 7 nt and 6 nt toeholds, respectively.

## The concentration of invader and anti-invader determines the kinetics of droplet dissolution and formation

The kinetics of droplet dissolution and formation should depend on the level of inhibiting (invader) and activating (anti-invader) molecules relative to the level of monomer (nanostar), as anticipated by our simple model (Fig. 1D). To quantify this dependence, we measured droplet areas over time after sequential addition of invader and anti-invader (AI), reported in the box plots in Fig. 2E. The area of individual droplets was normalized with respect to the average area of the sample prior to initiating invasion; the first measurement for anti-invasion experiments was taken 2 min after adding AI. The droplet average area, total area, and total number are compared across invader/AI levels in Fig. 2F; each quantity is normalized to the initial value in the sample prior to adding the invader. As the model predicts, the time it takes to dissolve droplets decreases monotonically with the invader to nanostar ratio, with 1× invader resulting in droplet dissolution within 5 min and 0.5× invader dissolving droplets in 15 min. A 0.25× level of invader only reduces the average droplet size without causing their complete dissolution. In contrast, the regrowth kinetics after the addition of AI do not monotonically scale with the amount of AI, as anticipated by our model (Fig. 1E). A 0.5× amount of AI (equimolar to 0.5× invader added) yields a faster regrowth of droplets. The 0.25× and 1× experiments produce a slower regrowth, albeit presumably for different reasons: at 0.25X, slow coarsening is likely the driving process, while at 1× the rapid activation of a large number of monomers likely makes nucleation of small droplets the dominant process yielding an overall smaller average size; the box plots of droplet data 5 min after adding AI confirm the predominance of small droplets in both cases (Fig. 1E, right). On samples invaded with 0.5× invader, we tested the effect of adding 2× excess AI (2× relative to nanostar level, 4× relative to invader) and we found that it does not accelerate droplet regrowth when compared to 0.5× AI; 4× AI excess actually decreases droplet regrowth (Supplementary Fig. 6). This can be explained by the fact that the AI sequence includes the 4-base palindromic domain of the nanostar sticky-ends, making it possible for the AI to associate with nanostars and prevent their separation, an effect that becomes predominant when AI is present in excess.

Given that droplets rapidly dissolve and reform upon the addition of 0.5× invader and AI, we asked whether their addition in multiple sequential cycles would yield similar results. Figure 2G shows six separate cycles of invasion and anti-invasion: invader and AI added in each cycle bind and form a fully double-stranded 'waste' species that accumulates over time but is expected to be inert. The incorporation of inert waste in the condensates may be the reason why cycles 5 and 6 show a more pronounced regrowth. Overall, these results show that programmable toehold-mediated strand invasion and displacement can be used to direct the self-assembly of large DNA condensates isothermally and reversibly by activating and deactivating the motifs participating in the condensation process.

## Valency reduction is a high-gain mechanism to control condensate kinetics

The observation that only 0.5× invader results in complete droplet dissolution prompted us to examine more carefully the influence of concentration of active DNA nanostars on their capacity to phase separate. Because high nanostar concentrations were adopted in previous work (typically above 5 μM), we tested whether condensation is at all possible at lower nanostar concentrations, with an expectation that no condensation would occur below 2.5 μM (the level of active

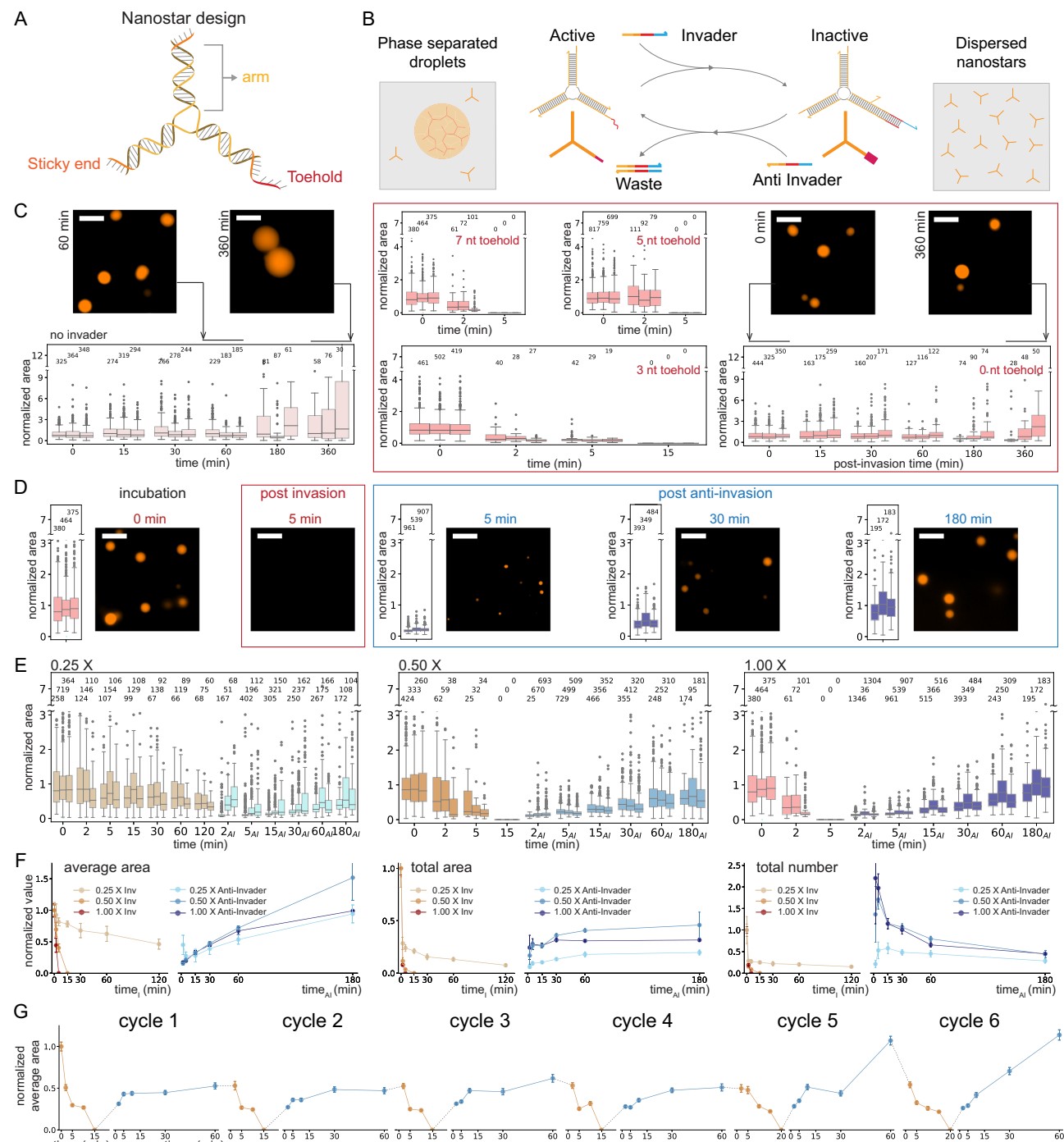

**Fig. 2 | Controlling condensation of DNA motifs over time via strand invasion and displacement reactions. A** Schematic of 3-arm DNA nanostar. One of the nanostar sticky ends was modified to include a single-stranded overhang or toehold (red domain on the 5′ end of one arm). **B** Schematic of the chemical reactions of invasion and anti-invasion for controlling the capacity of nanostars to yield phase-separated droplets. **C** Left: images and box plots illustrating droplet growth in the absence of invaders. Middle: box plots of droplet area after addition of invaders to nanostars with 7, 5, and 3 nt toeholds. Right: nanostars without a toehold do not dissolve after the addition of invaders. **D** Representative microscopy images of condensate droplets before the invasion, post-invasion (at 1×) and post-anti-invasion (at 1×), and box plots of condensate area. **E** Invasion and anti-invasion reactions for different concentrations of invader and anti-invader (from left to right: 0.25×, 0.5×, and 1×). For the anti-invasion experiments, the first measurement

was taken 2 min after the addition of anti-invader to the corresponding invaded sample. **F** Normalized average area, normalized total area, and total number of condensates during invasion and anti-invasion reactions for different concentrations of invader and anti-invader. **G** Invasion and anti-invasion reactions can be sequentially repeated multiple times (bound invader and anti-invader form an inert complex whose concentration increases during this experiment). **C**–**F** Results of n = 3 experimental repeats. In box plots, the central line indicates the median, and the bottom and top edges of the box indicate the 25th and 75th percentiles. Whiskers extend to the most extreme data points not considered outliers. Dots indicate outliers. The number of measured condensates is indicated above the respective box. **F** Error bars represent the standard deviation of the mean over three experimental repeats. **G** Results of a single experiment; error bars derived from bootstrapping. Scale bars are 10 μm.

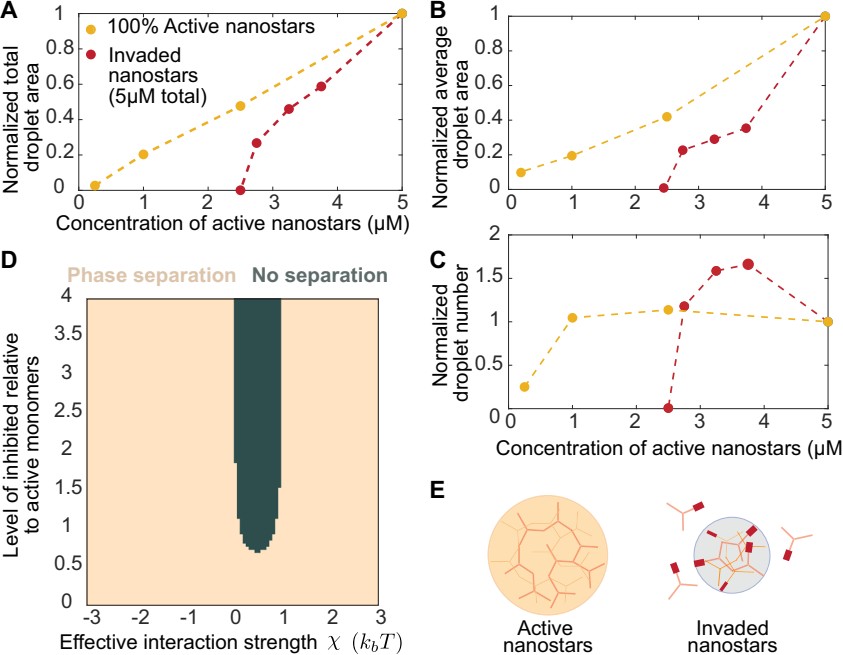

**Fig. 3 | Invasion allows for efficient control of phase separation. A** Normalized total condensate area was measured 30 min after annealing versus the level of active nanostars in the presence or absence of an invader. **B** Normalized average droplet area, and **C** Normalized average droplet number, measured 30 min after anneal. **D** Computational phase diagram (Supplementary Note 1) illustrating the existence of a range of interaction strengths between active and inactive monomers in which phase separation is suppressed at -0.6 ratio of inactive/active monomers. **E** Schematic representation of the invasion process causing a reduction of the valency of individual nanostars. Invaded nanostars can still interact with other active nanostars, limiting their capacity to form condensates. Normalization is done relative to the area (average or total) measured at 5 μM nanostar. Each data point is drawn from a single experiment monitoring droplet growth kinetics (Supplementary Figs. 8–16).

nanostars remaining after addition of 0.5× invader in our experiments). In contrast, we found that condensates form at concentrations as low as 0.25 μM.

To further investigate this, we monitored the formation of droplets when 5 μM DNA nanostars are annealed in the presence of an invader (0.25× to 0.5×). The droplet normalized total area, average area, and number 30 min after annealing are shown in Fig. 3A–C (dark red) versus the concentration of active nanostar, calculated as the difference between their total concentration and the concentration of invader. These graphs make it possible to compare two distinct series of experiments that have the same amount of active monomers, but one includes invaders, and the other does not. In the presence of an invader, condensation is completely suppressed by inactivating (valency reduction) only 1/2 the nanostar population. In the absence of an invader, a 1/20 reduction of nanostar concentration (nominal 5 μM) is required to suppress condensation. In other words, invasion increases the threshold of active nanostars needed for condensation. Further, a comparison between the slopes of the red and yellow curves in Fig. 3A indicates that the presence of an invader offers a 2× "gain" toward suppressing condensation. Overall, invasion appears to be an ultrasensitive mechanism because condensate formation is triggered or suppressed under small changes in the concentration of active monomers relative to the critical threshold. The fact that only half of the nanostar population requires deactivation for droplet dissolution prompts us to suggest a model in which inactive nanostars (valency two) interact with active ones, sequestering them and reducing the critical concentration for phase separation (Fig. 3E). In other words, a single invasion event affects multiple nanostars resulting in the observed "gain".

Using our previously developed theoretical model (Supplementary Note 1), we can understand how the addition of inactivated nanostars can affect the stability of droplets. While one might first expect that there should be no effect, a model where inactivated

nanostars and active nanostars interact with an interaction constant $X$ can lead to droplet dissolution, provided the parameter $X$ lies in a particular range. An $X < 0$ would imply that the inactivated nanostars are still attracted to being in the droplet phase and thus participate in phase separation with active nanostars. An $X > 2$ corresponds to strong exclusion from the droplet, leading to no inactivated nanostar inside the droplet. With simulations, we find that for $X$ between these values (in other words, the inactivated nanostars do not participate in phase separation but are not strongly excluded from the droplet), there is a decrease in effective droplet binding energy as the inactivated nanostars enter into the droplets. This alters the thermodynamic balance between dispersed and dense phases and can lead to droplet dissolution. Considering the molecular properties of active and inactive nanostars, it is likely that we are in this region of the phase diagram (and this is also observed experimentally). The inactivated nanostars have a valency of 2. Thus, they have some propensity to enter into existing droplets but cannot independently generate condensates at this valency[27].

These experimental and computational results are consistent with recent observations we made on a similar system in which 3 arm nanostars include one arm that is protected through a hairpin[28]. Like here, a critical ratio of active/inactive nanostars around 0.5 was required for condensate formation.

## Dissolution of droplets formed by monomers of variable size

Our theoretical model predicts that dissolution speed depends on the diffusion rate of the inhibitor molecule (Fig. 1D). To test whether we could change the diffusion of invaders into the condensate and thus droplet dissolution speed, we varied the length of nanostar arms between 8 and 24 base pairs (bp) (Fig. 4A). Because we considered arm lengths well below the persistence length of double-stranded DNA (50 nm, around 150 nt), it is reasonable to expect that longer arms yield droplets that are less densely packed and more permeable to invaders.

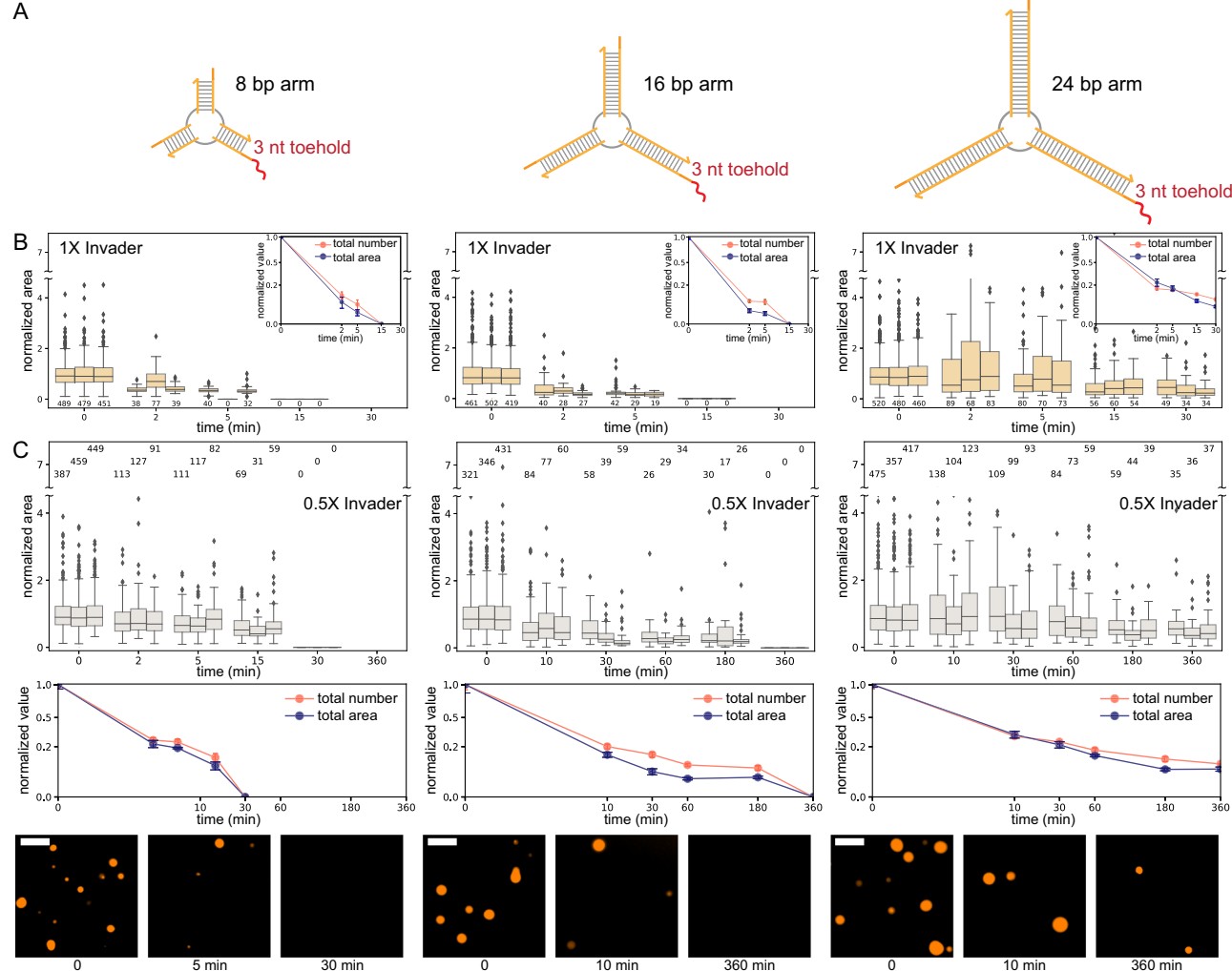

**Fig. 4 | Dissolution of droplets produced by nanostars of different sizes.**
**A** Schematic representation of the nanostars with different arm lengths, the same sticky-end domain, and a 3nt toehold. **B** Addition of 1× invader, normalized droplet area (box plot), number, and total area (inset). **C** Addition of 0.5× invader, normalized droplet area (top row), number, and total area (middle row). The bottom row provides representative images of the samples at the beginning and 5 and 60 min after the addition of the invader. The bottom row provides representative images of the samples at the beginning and 5 and 60 min after the addition of the invader. Results of $n = 3$ experimental repeats. In box plots, the central line indicates the median, and the bottom and top edges of the box mark the interquartile range. Whiskers extend to the most extreme data points not considered outliers. Dots indicate outliers. The number of measured condensates is above or below each box. In the total number and area plots, dots indicate the mean over the three experimental repeats, and error bars indicate the standard deviation of the mean. Scale bars are 10 μm.

We first verified that all variants form droplets in the absence and presence of a toehold. Because the average droplet size scales with the nanostar arm length[29], 8 bp arm nanostars yield noticeably smaller droplets than the 24 bp arm variant post anneal. However, the 24 bp arm design did not yield droplets in the presence of a 7 nt toehold (32 bp arm nanostars did not condense in the presence of a toehold of any length; Supplementary Fig. 2). For this reason, in all variants we adopted a 3 nt toehold, which in the nominal 16 bp arm design promoted droplet dissolution within 15 min at 1× invader (Fig. 2B, middle).

At 1× invader level, condensates dissolve within 15 min for the 8 bp and 16 bp variants but not for the 24 bp arm variant. Figure 4B reports box plots of droplet area normalized with respect to the initial average area, with insets showing the total area and number normalized with respect to their initial value. The droplet number sharply decreases within 2 min for all variants.

At 0.5× invader level, droplets in the 8 bp arm variant take about 30 min to dissolve, with total area and number gradually decreasing, as shown in Fig. 4C. For the 16 bp arm, it takes more than 3 h to dissolve

droplets. Droplets persist in the 24 bp arm variant, although the average and total condensate area decrease significantly.

These experiments do not show the expected correlation between nanostar size and invasion speed, suggesting that either the diffusion rate of invaders may not significantly differ in the size range we considered or that other effects come into play. For instance, our theoretical model predicts that the time it takes a droplet to disappear scales with their radius (Supplementary Fig. 20B), and on average the droplet radius is reduced by making the nanostars smaller[29]. In addition, by reducing the nanostar volume, we also increase the concentration of invadable sticky ends inside droplets, which should decrease the time required for dissolution. These effects suggest that, even though smaller nanostars should hinder invader diffusion, droplets can still dissolve fast, as is indeed observed in the experiment.

### Selective control of monomer valency via individually addressable toeholds

The valency of DNA nanostars is determined by the number of arms, a parameter that is known to affect their phase diagram[10,19,20,30]. Here

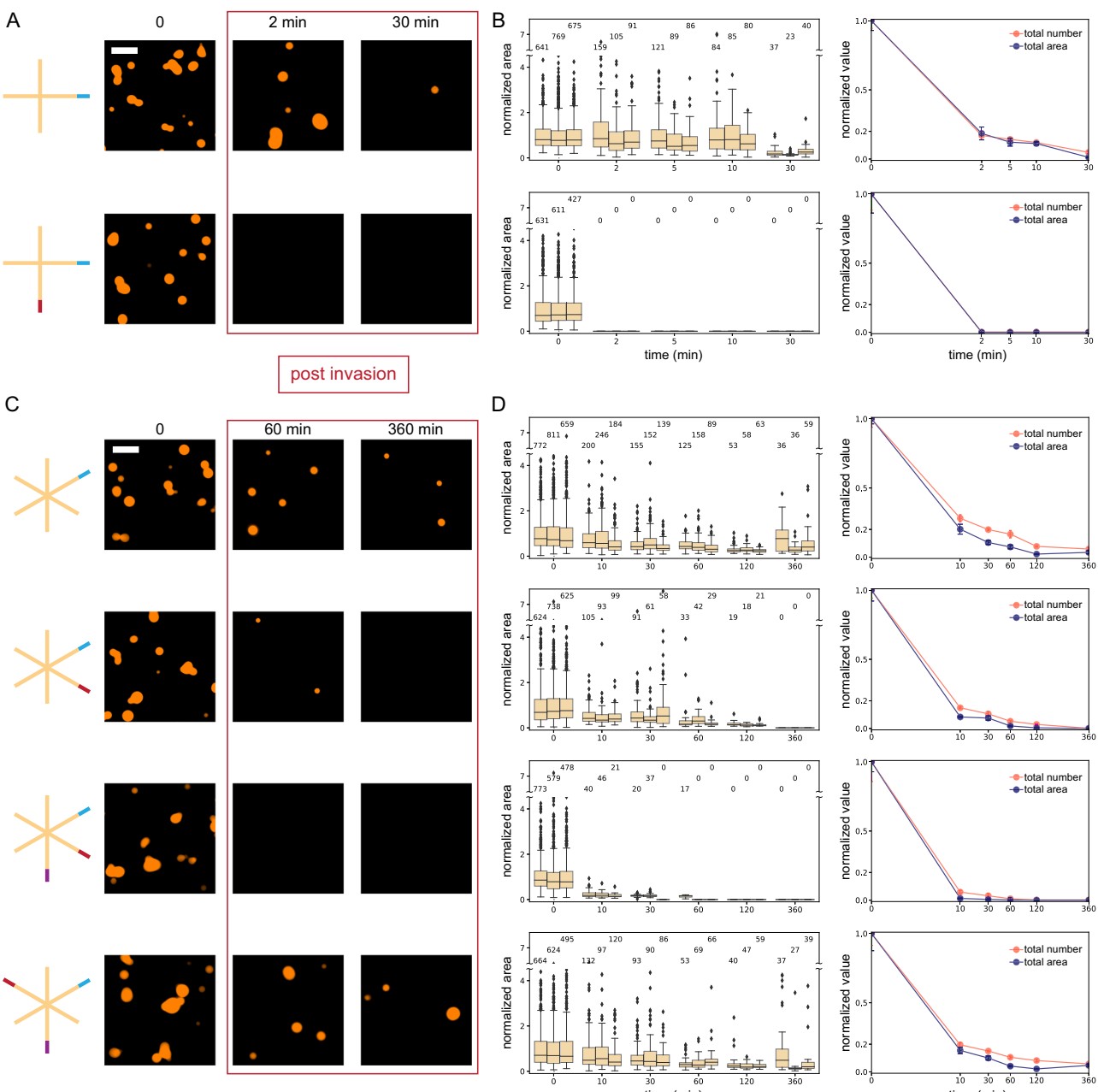

**Fig. 5 | Controlling the valency of nanostars via distinct toeholds. A** Schematic representation of the nanostars with 4 arms and one or two toeholds for invasion. Representative images show the change in the droplets after invader addition corresponding to each toehold region present on the variant (each invader = 1×). **B** Box plots of normalized droplet area (left) and total droplet area and number (right); in condensates formed by 4 arm nanostar with two invasion points, droplet area/number decrease more rapidly than those formed by 4 arm nanostar with one invasion point. **C** Schematic representation of the nanostars with 6 arms and one, two, or three toeholds for invasion. Three toeholds are placed next to each other in the adjacent variant and in an alternating pattern in the staggered variant. Microscopy images show the change in the droplets after invader addition corresponding to each toehold region present on the variant (each invader = 1×). **D** Normalized

droplet area (box plots, left), total area, and normalized droplet number (right). Droplets formed by 6-arm nanostars with one invasion point do not completely dissolve. With three adjacent invasion points, droplets dissolve within 120 min of invader addition. With three staggered invasion points, droplets behave similarly to the 6-arm nanostar with two invasion points. Results of *n* = 3 experimental repeats. In box plots, the central line indicates the median, and the bottom and top edges of the box mark the interquartile range. Whiskers extend to the most extreme data points not considered outliers. Dots indicate outliers. In the total number and area plots, dots indicate the mean over the three experimental repeats, and error bars indicate the standard deviation of the mean. Condensate numbers are reported above each box. Scale bars are 10 μm.

we demonstrate that, like three-arm nanostars considered so far, nanostars with higher valency can be controlled by strand invasion reactions that target specific toehold domains. Further, we investigate the kinetics of droplet dissolution as valency is progressively reduced by invader binding.

Nanostars with 4 and 6 arms, presenting distinct sequences in each arm but identical palindromic 4 nt sticky-ends, were modified to

include a variable number of 7 nt toeholds, each characterized by a distinct sequence (schematics in Fig. 5A, C). We then designed a set of invaders, each complementary to a single specific toehold and arm, so that individual arms on a nanostar can be selectively deactivated. For these experiments, we used stoichiometric amounts (1×) of each invader with respect to the nanostar level (5 μM), to deactivate all the corresponding sticky-ends and fully reduce the motif valency.

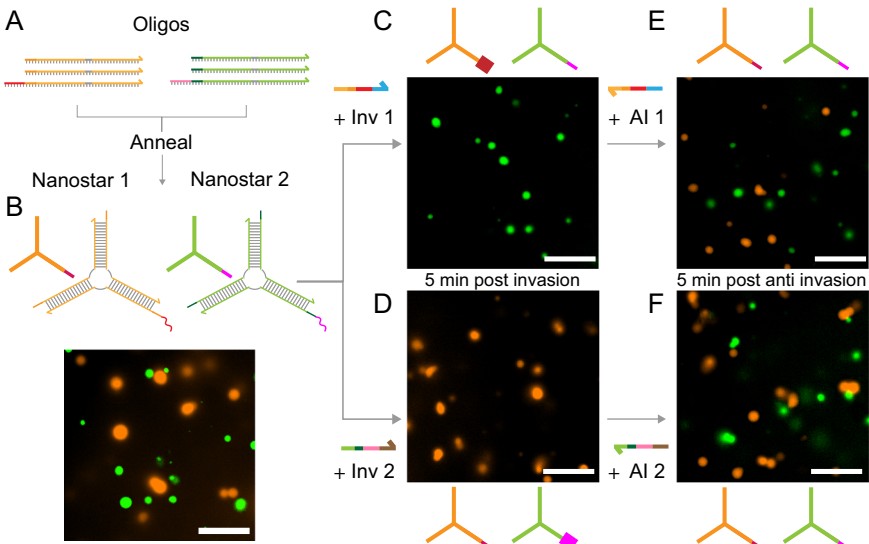

**Fig. 6 | Sequence specific dissolution and regrowth of distinct condensates.**
**A**, **B** Orthogonal DNA strands can be annealed to form orthogonal DNA nanostars that do not interact. **B** Nanostar 1, labeled with Cy3, is represented in orange and nanostar 2, labeled with FAM, is represented in green. **C**, **D** Invaders specific to either Nanostar 1 or Nanostar 2 selectively inhibit their capacity to phase separate while leaving the other unaffected, as shown by microscopy images. **E**, **F** Respective Anti-invaders for each design reactivate their capacity to phase separate again. Microscopy images were taken 5 min after the addition of the respective invader or anti-invader strands. Representative images of two experimental replicates. Invader and AI = 1×. Scale bars are 10 μm.

For the 4-arm motif (Fig. 5A, B), we found that the invasion of a single sticky-end is sufficient to dissolve droplets within 30 min, causing an 80% drop in the total area and the number of droplets within 2 min. The invasion of two sticky ends accelerates dissolution, which is complete in under 5 min (Fig. 5A, B). Removal of invaders by their specific anti-invader results in droplet regrowth (Supplementary Fig. 7).

For the 6-arm motif, invasion of a single sticky end is insufficient to completely dissolve droplets. However, their number and total area are reduced by 70–80% within 10 min. Invasion of two sticky ends reduces the droplet number by more than 80% in five minutes, but the droplet area still decreases slowly, and a few droplets persist for several hours (Fig. 5C, D). Invasion of 3 adjacent sticky-ends is necessary to completely dissolve the droplets, with over 90% of droplets dissolved in the first 10 min. By invading three staggered sticky-ends yields we observe a droplet dissolution behavior comparable to the invasion of two adjacent sticky-ends. After 24 h, the staggered three-arm design shows very large droplets, see Supplementary Fig. 3. These results suggest that the orientational order of the nanostar arms is important for condensation, meaning that if the active arms cannot easily connect in a tessellating pattern, formation of condensates becomes more difficult. These effects could be further explored by resorting to coarse-grained simulations[31] or by additional experiments systematically varying arm length.

It is particularly surprising that deactivation of only 1 or 2 out of 6 arms still significantly reduces the condensed mass, even though these nanostars are not expected to be near the phase separation boundary (they have a higher melting temperature than their 3 arm counterpart, which assembles robustly at concentrations ranging from 0.25 to 5 μM as shown in Fig. 3A). We expect that, although invaders here were designed to address specific toeholds and arms, they can still weakly interact with the palindromic sticky-ends of any arm: this should contribute to arresting coarsening and potentially slowing down coalescence as we observed in non-toehold 3 arm nanostars supplied with invader (Fig. 2C, right plot). In addition, invaders bound to the sticky ends could reduce or eliminate attractive forces among nanostars while increasing steric and electrostatic repulsion. Droplet dissolution in the case of 1 and 2 arm invasion may be due to the combined effects of toehold-mediated invasion, competitive binding of invaders to sticky ends, and increased repulsion among nanostars.

## Programmable dynamic control of dissolution and formation of distinct coexisting condensates

A major advantage of DNA nanotechnology is the possibility to build motifs and assemblies that are structurally identical but operate as orthogonal, non-interacting species. This principle can be extended to the design of motifs yielding condensates[20,32], and to the reactions that control them. To illustrate this advantage, we designed two orthogonal 3-arm DNA nanostars with different sticky ends and labeled them with distinct fluorophores (Nanostar 1, Cy3; and Nanostar 2, FAM; shown in Fig. 6). As DNA backbones are negatively charged, DNA nanostars do not spontaneously interact in the absence of base-pairing. To minimize the interactions between palindromic SEs of the two designs, nanostars were designed to have 6 nt sticky ends. (Notably, palindromic 4 nt sticky-ends containing only 'A's and 'T's do not yield condensates, see also Supplementary Fig. 4). The use of 6 nt sticky-ends, however, is expected to raise the temperature at which the condensates transition from a liquid to a gel state as noted in the literature[20,33]. This is confirmed by the fact that even though invasion yields droplet dissolution at room temperature within a few minutes, we were unable to regrow these condensates even after 3 h of anti-invader addition (Supplementary Fig. 5). By raising the temperature to 34 °C, we were able to recover reversible phase transitions using invader and anti-invader.

The two nanostar types were annealed in the same pot, producing condensates of distinct colors, orange for Nanostar 1 (Cy3) and green for Nanostar 2 (FAM). The droplets remained demixed and were not observed to fuse (Fig. 6B). We then split the sample in two: we added sequentially the invader (1×) and the anti-invader (1×) designed for Nanostar 1 to the first aliquot, observing complete dissolution and then regrowth of the orange droplets, while the distribution of green droplets is qualitatively unaffected (Fig. 6C, E). Similarly, when invader and anti-invader for Nanostar 2 were sequentially added, we observed dissolution and regrowth of the green droplets, while the orange ones remained intact (Fig. 6D, F).

These results demonstrate the design of two orthogonal artificial DNA condensates that are dynamically controllable via tailored chemical reactions, and we expect they can be easily scaled to systems in

which dozens of distinct condensates are individually addressable by sequence-specific DNA or even RNA regulators.

## Discussion

All forms of life require the presence of physically separated compartments that isolate molecules and reactions. Similarly, artificial materials may achieve life-like behaviors by combining the operation of distinct functional partitions. In both contexts, condensation is a useful approach to self-organize molecules in the absence of membranes[21,34]. An important characteristic of condensates is that they form and dissolve dynamically, shuttling guest molecules across distinct phases. Learning how to control these dynamic properties via biochemical reactions under homeostatic conditions will make it possible to build complex molecular systems and materials that self-organize in space and time, provide insights into similar phenomena in biological cells, and possibly offer hypotheses on how life originated[35].

Through theory and experiments, we have defined a model problem to elucidate how chemical reactions can achieve dynamic control over condensation. We have considered a phase separating species that is sequentially inhibited or activated via reactions that turn on or off its ability to condense. First, we have set up a computational model that sets up expectations for the thermodynamic and kinetic behavior of condensate droplets under different concentrations of inhibitor and activator, reaction rate constants, and diffusion rates. Next, we have realized this model problem through a platform of DNA components that implement both a phase-separating process and the chemical reactions that regulate it. We achieved this by engineering DNA nanostars, a versatile motif that generates phase-separated condensates depending on the number of arms and on the sequence of their sticky ends, which determine the valency and bond stability of the nanostars[19,20]. Nanostars were modified to include toeholds that allow for sequence-specific strand invasion of individual arms by invader DNA molecules, making it possible to selectively reduce their valency, causing droplet dissolution under a variety of conditions. Displacement of the invader via an anti-invader strand results in reactivation of the nanostars and regrowth of droplets. An alternative method for controlling DNA condensate dissolution is to displace linker strands[36], although these studies provided limited characterization of the design parameters involved. We note that separated droplets may experience transitions between liquid and gel that could be studied via rheology experiments falling outside of the scope of this paper.

We have shown that condensates can be regulated by chemical inputs that change the balance of the phases leading to shrinking or growth. Following on this, we can broadly identify two different mechanisms of control over condensation through inhibitors, which we deem *thermodynamic* and *chemical*. In thermodynamic control, a system close to the transition point can be pushed across the phase separation boundary, leading to a region in phase space where the dispersed phase is favored. This could proceed exclusively via sequestration of monomers in the dispersed phase, leading to a thermodynamic phase transition where the droplets slowly lose material through thermal fluctuation. Chemical control, by contrast, proceeds via the deactivation of the monomers in both the dispersed and the dense phase, as monomers participate in chemical reactions. While our experiments focused on establishing means for chemical control, the response of our system is due to the simultaneous presence of thermodynamic and chemical control. For example, Fig. 3 shows that an amount of inhibitor that is insufficient to deactivate all the monomers can still lead to complete dissolution of the droplets because a mix of activated and deactivated monomers presents a different phase diagram when compared to active monomers alone.

The dynamic condensates we built using DNA nanotechnology show unexpected behaviors that may be generalizable to other biomolecular condensates. First, we have found that valency reduction induces complete or partial condensate dissolution in all cases considered. In trivalent nanostars, this reaction has an effect comparable to a sharp increase of the critical concentration threshold for condensation, indicating that valency reduction is an efficient mechanism to control phase separation. This finding may be significant biologically, as it points to the possibility that the size and number of biomolecular condensates may be controlled rapidly by regulators that interact with a fraction of phase-separating monomers, whose total mass may be conserved. This is reminiscent of the operation of phosphorylation pathways, which transmit signals by activating and deactivating their protein targets on timescales much faster than protein production and degradation. We also observed that the interactions among invaders and nanostars can, in some cases, stabilize the condensate size and number, a yet-to-be-explained phenomenon that may be due to a combination of factors, including changes in the condensate surface[37]. Finally, through engineering nanostars with different placements of invasion toeholds, we demonstrated that the orientation of the remaining active arms plays a significant role in the formation of condensates. This information could be used to customize DNA condensates not only to achieve prescribed thermodynamic and kinetic properties but also to introduce domains that harbor particular functions, like recruiting guest molecules or performing catalytic reactions[38,39].

Our work builds on many recent demonstrations of responsive DNA-based soft materials. Entangled DNA strands, linear cross-linked motifs, and nanostars have been engineered to generate hydrogels with tunable mechanical and rheological properties[19,40]. Dynamic responses in DNA hydrogels have been obtained by using strand displacement, for example, to control DNA hydrogel stiffness[41], to program swelling[42], and to introduce re-entrant behaviors[43]. Further, by integrating specific DNA motifs, DNA-based hydrogels can be made responsive to a variety of chemical and physical stimuli, including enzymatic reactions, pH, light, and temperature[40]. Similarly, DNA condensates may be designed to respond dynamically to many types of inputs, as recently illustrated by studies of their enzymatic degradation[44] and light-mediated activation[28]. Orthogonal sequence design has been used previously to build condensates that coexist without mixing[20,32], a behavior that we were able to reproduce here. These results may be systematically expanded to produce libraries of DNA or RNA condensates that can be selectively grown and dissolved.

The dynamic condensates described here inherit the programmability of systems previously demonstrated by DNA nanotechnology[9] and could be coupled with a variety of other DNA devices. DNA condensates could be designed to respond to multiple inputs and complex chemical reactions through the expanding number of modular sensors, logic circuits, and dynamic circuits based on strand displacement[13]. Molecular instructions may even include algorithmic or dynamic behaviors, as demonstrated in crystalline DNA structures[16,45,46]. Finally, responsive nucleic acid condensates like those described here may be used as scaffolds to other materials[36,47] or to localize molecules and reactions on demand[48,49], expanding the toolkit of DNA and RNA materials for the advancement of artificial life, synthetic biology, and biotechnology.

## Methods

### Oligonucleotides

Sequences are listed in the Supplementary Tables 1–15. Oligonucleotides were purchased from IDT DNA. Fluorophore-labeled strands were purified to high-performance liquid chromatography (HPLC) grade. Strands of more than 60 bases in length were purified by polyacrylamide gel electrophoresis (PAGE). All other strands were ordered standard desalted.

### Sample preparation

All the motifs were formed by mixing the desired concentration of each component oligomer in a buffer consisting of 20 mM Tris-HCl

(pH 8.0) and 350 mM NaCl. Samples were prepared using LoBind Eppendorf tubes. One of the nanostar strands was modified to include a fluorescent dye, which was mixed at a 2.5% molar ratio in the solution for Cy3, and a 10% molar ratio for FAM. Nanostars were annealed by placing samples in a thermocycler, in which they were heated at 95 °C for 5 min and then cooled to room temperature at a rate of −1 °C/min. At the end of annealing, condensates were allowed to grow at room temperature (set at 27 °C in an incubator) unless otherwise specified.

Invaders were added to assembled nanostar samples 30 min after the end of the annealing process. Anti-invaders were added after variable incubation times with the invader, as noted in the corresponding figures. Aliquots for imaging were drawn from the solution, including nanostars, invaders, and anti-invaders at the times specified in each figure. Samples were rapidly imaged using observation chambers described below. Aliquots were drawn consistently from the center of the test tube containing the sample; prior to taking an aliquot, the sample was gently mixed by swirling the pipet tip, with the goal of limiting the effect of sedimentation on the droplet size distribution.

### Sample imaging

Samples were imaged using an inverted microscope (Nikon Eclipse TI-E) with Nikon CFI Plan Apo Lambda 60× Oil (MRD01605) objective. Cy3 and FAM were visualized at excitation wavelengths of 559 nm and 488 nm, respectively, using Semrock BrightLine Epi Filter cubes. Samples were imaged using coverslips (Fisherbrand™, cat: 12-545-JP) measuring 60 × 22 mm, with a thickness between 0.13 and 0.17 mm, were soaked in 5% (w/v) bovine serum albumin (BSA) and dissolved in 20 mM Tris-HCl (pH 8.0) for over 30 min to prevent nonspecific interactions of the DNA on the glass surface, and avoid wetting and surface-induced nucleation of condensates. After the BSA coating, the glass slides were washed twice with distilled water and dried under an airflow. A square parafilm (Parafilm M® from Fisher Scientific, cat: S37440) slice with a punched hole in the middle was stuck to the BSA-coated glasses by heating the slides to 50 °C for 1 min before imaging. After the coverslips returned to room temperature, 2.5 μL of sample was pipetted out in the punched hole. Another smaller coverslip (Fisherbrand™, cat: 12-545-AP) measuring 30 × 22 mm, with a thickness between 0.13 and 0.17 mm, was placed on top of the parafilm to avoid evaporation of the sample solution during the observation period. The BSA coating reduces the wetting of the glass and makes it possible to observe nearly spherical droplets with negligible surface interaction during our typical measurement time.

### Quantification and statistical analysis

We extracted DNA condensate size, number, and eccentricity measurements from epifluorescence micrographs using a custom Python script available on Github: https://github.com/klockemel/Condensate-Detection.

This script implements several Python packages, including scikit-image, pandas, and others[50–52].

Unless otherwise noted, averages and standard error of the mean (SEM) for condensate measurements were generated via triplicate experimental replicates, including hundreds of droplets.

Box plots were generated using in-house Python scripts integrated with the seaborn library. Epifluorescence images capture objects both inside and outside of the plane of focus. Objects outside of the plane of focus may be a different size or shape than they appear in an image and have softer edges than objects in the plane of focus. To maintain confidence in our measurements of condensate characteristics, we sought to measure only objects within the plane of focus by implementing an edge-based threshold method. First, each image is smoothed with a Gaussian filter to limit the influence of noise inherent to a fluorescence micrograph in the detection of condensates. A Sobel filter is then applied to the smoothed image to find the edges within the image. An Otsu threshold is used to separate condensate edges from the background of the image, followed by binary operations to clean the resulting binary image of edges. The image is thinned such that each feature or edge is 1 pixel thick. Enclosed edges are then filled with a binary fill holes method. A binary opening of the image removes any unenclosed regions, such as lines or speckles. Finally, the image is dilated using a disk of radius 5 pixels. Without the dilation, objects larger than about 1.5 μm in diameter are systematically underestimated in the threshold process. As a majority of the condensates observed in this work are larger than 1.5 μm in diameter, we chose to include the dilation, although it systematically overestimates small objects. Finally, the area, diameter, and eccentricity of individual condensates are measured, as well as the total number of condensates. Diameter is estimated as the diameter of a circle with the same area as a given condensate. All user-input parameters for each image are saved in a CSV file, and a diagnostic image with labeled condensates is generated. We processed 8 images (identical size of the field of view) for each time point and condition. Within a set of 8 images, if there were less than 16 condensates with either an average diameter less than or equal to 1.5 μm or a total area less than 28.32 μm² (which is the sum area of 16 1.5 μm diameter condensates) we considered the amount of condensate for that condition to be 0.

**Normalization.** We normalized droplet areas reported in box plots by dividing each droplet area by the average droplet area of the sample prior to adding invader or anti-invader. The normalized total area of droplets at each time point was computed by dividing by the total area prior to adding the invader or anti-invader. Similarly, the droplet number was normalized by dividing it by the number of droplets measured in the sample prior to adding invader or anti-invader.

**Averages and error bars.** Unless otherwise noted, the kinetic condensate data we report are from three experimental replicates. Error bars represent the standard deviation of the mean.

**Bootstrapping.** Applies only to Fig. 2G in the main paper and to the data in Supplementary Figs. 8–16. To bootstrap the data, three sub-samples were generated by grabbing a random assortment of half the observations at each time and condition. The normalized average area was calculated for each of the three sub-samples as described above, and the average and SEM were calculated for these three values. We compared the performance of our bootstrapping method with SEM computed through MATLAB's bootstrap function, finding similar predicted variability.

### Reporting summary

Further information on research design is available in the Nature Portfolio Reporting Summary linked to this article.

## Data availability

Source data for figures are provided in the paper. Source data are provided in this paper.

## Code availability

The Python code we developed for image processing is available on GitHub. The Mathematica code developed for simulating condensate kinetics and phase diagrams is available on Zenodo.

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

## Acknowledgements
This work is dedicated to the memory of Melissa A. Klocke. This research was supported by the US National Science Foundation through CAREER award 1938194 and FMRG: Bio award 2134772 to E.F., and by the Alfred P. Sloan Foundation through award G-2021-16831. We thank Deborah Fygenson, Paul Rothemund, Eli Kengmana, and Rebecca Schulman for their helpful advice.

## Author contributions
S.A., D.O. and E.F. designed research. S.A. and M.D. performed experiments and analyzed data. E.F. analyzed data. D.O. designed and conducted computational simulations. M.A.K. developed image processing methods and analyzed data. All authors contributed to writing the paper.

## Competing interests
The authors declare no competing interests.
