## [Peer Review File · Nature Communications]

Editorial Note: This manuscript has been previously reviewed at another journal that is not operating a transparent peer review scheme. This document only contains reviewer comments and rebuttal letters for versions considered at *Nature Communication*.

REVIEWER COMMENTS

Reviewer #1 (Remarks to the Author):

I do not understand the history of this article, and how it ended up being submitted for revision 8 months after the last time I saw it. The implication in the rebuttal seems to be that it was tied up in the journal's hands for some time, which (understandably) complicates the author's ability to revise and improve the manuscript, given personnel issues. The journal staff might consider the pressure being put on the authors by these sorts of actions.

That said, my assessment is that the article is improved in some, but not enough, respects. The good points are that the article is pared down and so a bit more coherent and focused. The theory is a bit better integrated. The article still has a very compelling motivation; particularly the introduction is well written, and I remain of the opinion that the experiments performed are potentially significant. Dynamic control of condensates is a very compelling area to study.

However I ultimately find the improvements in the revision to be modest, and many of the prior issues remain: Each figure introduces an intriguing experiment, but in many cases the actual data is somewhat unclear, and so doesn't obviously support the conclusions. A specific example of this is in Figure 5, where the authors carry out a very interesting experiment that tests the dependence of dissolution on the pattern of targeted arms in a 6-armed nanostar (i.e. 3 adjacent arms, vs 3 alternating arms). The claim in the main text is that there is a clear difference between the two designs, and it is speculated that this has to do with the differing ability of the two patterns to form meshworks. Yet the actual data does not support a significant difference in the dissolution trajectories of the two situations—to my eye, the bottom two plots in Fig 5 D are nearly the same, whether one looks at the box/whisker plots, or the line plots of area and number.

Generally, I just can't see that the experiments were carried out with enough rigor to support many of the claims. I have two specific fundamental concerns, that apply to nearly all the data shown:

- 1) The new methods section indicates that droplets were prepared in a test tube, and aliquots were withdrawn at various intervals to test the time course behavior through imaging. Yet, these droplets are heavier than water and so should sediment; after 10 or 30 minutes, I would expect most if not all the droplets to accumulate at the bottom of the tube, meaning that what is seen in the microscope could

heavily depend on exactly how the aliquot was withdrawn (was the pipette tip inserted into the sediment, or kept high up in the supernatant? etc). This is the sort of protocol dependence I was concerned about in my initial review, and the new methods section, while helpful, doesn't remove this concern.

2) It doesn't appear that appropriate replicates were carried out. From the text, it does seem that a control experiment just on phase separating droplets (without dynamic control) was performed in triplicate, and leads to estimates of coefficients of variation given in the middle of page 5; while I am surprised at the extraordinarily small CV quoted for, e.g., droplet number, I do think triplicate experiments are indeed sufficient. But then there is a statement in the methods that 'unless otherwise noted' quantitative data comes from a 'single experimental replicate'. I am thus lead to believe that apart from the control experiment, each experiment was carried out just once, which I do not find acceptable. I emphasize that this problem can't be solved just by bootstrapping that single replicate to get an error estimate. Any number of things could go wrong in one replicate (pipetting errors, undesired temperature jumps, etc) that could lead to systematic errors that contaminate the data set.

Ultimately I find this paper to be a difficult case—there are some very interesting and compelling ideas and experiments, and some tantalizing results, but things just don't seem to have been done carefully enough throughout.

We are grateful for the constructive comments of the Reviewer. We appreciate the concerns about the protocol robustness and about the need for reproducibility, which were communicated to us in a clear, thoughtful, and professional manner.

I do not understand the history of this article, and how it ended up being submitted for revision 8 months after the last time I saw it. The implication in the rebuttal seems to be that it was tied up in the journal's hands for some time, which (understandably) complicates the author's ability to revise and improve the manuscript, given personnel issues. The journal staff might consider the pressure being put on the authors by these sorts of actions.

[Redacted]

That said, my assessment is that the article is improved in some, but not enough, respects. The good points are that the article is pared down and so a bit more coherent and focused. The theory is a bit better integrated. The article still has a very compelling motivation; particularly the introduction is well written, and I remain of the opinion that the experiments performed are potentially significant. Dynamic control of condensates is a very compelling area to study.

We thank the Reviewer for the positive comments.

However I ultimately find the improvements in the revision to be modest, and many of the prior issues remain: Each figure introduces an intriguing experiment, but in many cases the actual data is somewhat unclear, and so doesn't obviously support the conclusions. A specific example of this is in Figure 5, where the authors carry out a very interesting experiment that tests the dependence of dissolution on the pattern of targeted arms in a 6-armed nanostar (i.e. 3 adjacent arms, vs 3 alternating arms). The claim in the main text is that there is a clear difference between the two designs, and it is speculated that this has to do with the differing ability of the two patterns to form meshworks. Yet the actual data does not support a significant difference in the dissolution trajectories of the two situations—to my eye, the bottom two plots in Fig 5 D are nearly the same, whether one looks at the box/whisker plots, or the line plots of area and number.

Generally, I just can't see that the experiments were carried out with enough rigor to support many of the claims. I have two specific fundamental concerns, that apply to nearly all the data shown:

1) The new methods section indicates that droplets were prepared in a test tube, and aliquots were withdrawn at various intervals to test the time course behavior through imaging. Yet, these droplets are heavier than water and so should sediment; after 10 or 30 minutes, I would expect most if not all the droplets to accumulate at the bottom of the tube, meaning that what is seen in the microscope could heavily depend on exactly how the aliquot was withdrawn (was the pipette tip inserted into the sediment, or kept high up in the supernatant? etc). This is the sort of protocol dependence I was concerned about in my initial review, and the new methods section, while helpful, doesn't remove this concern.

We acknowledge that sedimentation of droplets can pose a challenge affecting the experimental measurements. Being aware of this issue, we consistently mix the sample prior to taking aliquots for imaging, however we neglected to mention this in our previous versions of the manuscript. To address

this concern, we have clarified (in the manuscript and in the SI) that we gently mix the sample before withdrawing aliquots for imaging. This mixing step aims to allow for sampling droplets of all sizes, reducing the effect of sedimentation of larger droplets. We are aware that other groups store DNA condensate samples in a rotator to limit sedimentation; we decided against this, because we expect that rotation will introduce other biases and sources of uncertainty in the outcome of experiments monitoring droplet growth kinetics.

Further, we have conducted triplicate experiments for the majority of the data sets in the manuscript, which gives a measure of the impact of the sampling protocol on our observations. We found that our main conclusions remained consistent across different repeats with minor changes when compared to the data we originally reported.

2) It doesn't appear that appropriate replicates were carried out. From the text, it does seem that a control experiment just on phase separating droplets (without dynamic control) was performed in triplicate, and leads to estimates of coefficients of variation given in the middle of page 5; while I am surprised at the extraordinarily small CV quoted for, e.g., droplet number, I do think triplicate experiments are indeed sufficient. But then there is a statement in the methods that 'unless otherwise noted' quantitative data comes from a 'single experimental replicate'. I am thus lead to believe that apart from the control experiment, each experiment was carried out just once, which I do not find acceptable. I emphasize that this problem can't be solved just by bootstrapping that single replicate to get an error estimate. Any number of things could go wrong in one replicate (pipetting errors, undesired temperature jumps, etc) that could lead to systematic errors that contaminate the data set.

We now report three experimental replicates for the majority of our assays. The new experiments were almost entirely conducted by new co-author Mr. Dizani, as other team members became unavailable. This corroborates our confidence in the reproducibility of the results at the hands of different experimentalists.

New triplicate repeats were done for assays reported in Fig. 2C-F, Fig. 4, and Fig. 5, which were completely revised. In the revised figures we report statistical information about droplet area in each individual repeat through box plots, as well as total droplet area and number. For this reason, we eliminated the majority of SI figures, which included the same information in the previous version of the manuscript. As a consequence, the SI is now significantly shorter.

The text has accordingly been updated, although the trends we observed are consistent with what we reported prior. In particular, following on the main comment of the Reviewer, we confirm that adjacent vs staggered arm invasion for 6 arm nanostars does result in a noticeable difference in dissolution efficiency.

Minor changes in trends that we observed are: in Fig. 2C, right, we confirmed that with zero nt toehold, invaders arrest growth when compared to the no-invader control in Fig. 2C, left; however, we found a decrease in droplet number that wasn't noted in our previous experiments.

Fig. 4C, 16 bp arms, 0.5 invaders, the droplet area shown in the box plot, as well as the total area, decrease over time more markedly than in the previous round of experiments.

The following figures report the same experimental data included in the previous version of the manuscript:

Fig. 2G. We consider that each cycle is per se a technical repeat of invasion/anti-invasion addition.

Fig. 3, which includes the results of experiments in which nanostars are annealed in the presence or absence of invader, and imaged after annealing is complete. Here no DNA is added to the sample after annealing. Only one experimental parameter changes across experiments: the concentration of nanostars (yellow), or the concentration of invader (red) which is determined prior to annealing. The full kinetics of droplet growth are reported in SI Sections 3.8, 3.9.

Fig. 6; these experiments were originally replicated twice (the caption now mentions this), and only report qualitative information.

We opted against replicating experiments previously reported in the old SI Section 3.7; these data have now been excluded from the manuscript. Our choice is due to time constraints, and because these experiments are of secondary importance relative to the main contributions of the manuscript.

Ultimately I find this paper to be a difficult case—there are some very interesting and compelling ideas and experiments, and some tantalizing results, but things just don't seem to have been done carefully enough throughout.

We appreciate the Reviewer's feedback. We agree that replicating our experiments was important to support the claims made in the manuscript.

REVIEWERS' COMMENTS

Reviewer #1 (Remarks to the Author):

I appreciate the authors careful revision of the manuscript, including removing certain poorly-supported results, and, particularly, increasing the number of experimental replicates. This latter improvement removes my major critique of the article. So, based on my opinion of the importance of the subject of the article (expressed in prior reviews), and with a now sufficient experimental data set, I support publication of the article.